# Remediation Approaches to Reduce Hydrocarbon Contamination in Petroleum-Polluted Soil

**DOI:** 10.3390/microorganisms11102577

**Published:** 2023-10-17

**Authors:** Abdelkareem Elgazali, Hakima Althalb, Izzeddin Elmusrati, Hasna M. Ahmed, Ibrahim M. Banat

**Affiliations:** 1Faculty of Arts and Sciences, Tocra Campus, University of Benghazi, Benghazi P.O. Box 1308, Libya; abdelkarem09@gmail.com; 2Environmental and Biological Chemistry Research Center (EBCRC), University of Benghazi, Tocra P.O. Box. 9480, Libya; 3Environmental Research Department, Petroleum Research Center, Tripoli P.O. Box 6431, Libya; i.elmusrati@prclibya.com (I.E.); h.ahmed@prclibya.com (H.M.A.); 4Faculty of Life and Health Sciences, Ulster University, Coleraine BT52 1SA, Northern Ireland, UK; im.banat@ulster.ac.uk

**Keywords:** bioremediation, heavy metals, oil spills, contaminated soil, total petroleum hydrocarbon (TPH), microorganisms

## Abstract

Heavy metals pollution associated with oil spills has become a major concern worldwide. It is essential to break down these contaminants in the environment. In the environment, microbes have been used to detoxify and transform hazardous components. The process can function naturally or can be enhanced by adding nutrients, electron acceptors, or other factors. This study investigates some factors affecting hydrocarbon remediation technologies/approaches. Combinations of biological, chemical, and eco-toxicological techniques are used for this process while monitoring the efficacy of bacterial products and nutrient amendments to stimulate the biotransformation of contaminated soil. Different hydrocarbon removal levels were observed with bacterial augmentation (*Beta proteobacterium* and *Rhodococcus ruber*), exhibiting a total petroleum hydrocarbon (TPH) reduction of 61%, which was further improved to a 73% reduction using bacterial augmentation combined with nutrient amendment (nitrogen, potassium, and phosphorus). A heavy metal analysis of the polluted soils showed that the combination of nutrient and bacterial augmentation resulted in a significant reduction (*p*-value < 0.05) in lead, zinc, and barium. Toxicity testing also showed that a reduction of up to 50% was achieved using these remediation approaches.

## 1. Introduction

Oil and gas exploration and production threaten the physical environment, health, culture and economic and social structure of local communities through potentially severe environmental deprivation. In recent years, an increase in heavy metals has been detected in contaminated soil with crude oil and its refined products; in addition, the levels of certain hydrocarbons in the soil environment increase the risk of bioaccumulation of these toxic compounds in ecosystems that ultimately threaten human health. Heavy metals are toxic to microbial metabolic activities, which are essential to transform and detoxify heavy metals in the environment, especially in soils, to protect human health and to prevent bioaccumulation in the ecosystem [1]. Specific heavy metal toxicity has been investigated and findings reveal that copper, cadmium, lead, and zinc disrupt cellular functions, inhibit enzyme activities, and damage nucleic acids [2].

The difference between ubiquitous metals in the environment and other toxicants is that metals cannot be destroyed or created. Metals are incorporated in organic compounds or inorganic salts and their distribution in the environment occurs naturally via geological activity and biological cycles [3]. Heavy metals are toxic in their chemically combined forms: some are extremely toxic in most forms (such as mercury), and others are toxic in their elemental form. The temperature, pH, reducing nature (as expressed by the negative log of the electron activity), the nature of the solids, especially the inorganic and organic chemical functional groups on the surfaces, the cation-exchange capacities, and the surface areas of the solids largely determine the attenuation of heavy metal ions [4]. 

Heavy metals may undergo oxidation–reduction processes and precipitate as slightly soluble solids (especially sulfides) in addition to their ability of being sorbed and to undergo ion exchange with geospheric solids, and in some cases, undergo microbial methylation reactions that produce mobile organometallic species such as in the case of mercury [5]. In some instances, a chelating agent may be essential to treat the toxic effects of heavy metals. Soil is susceptible to heavy metal accumulation in the environment due to its composition of a variable mixture of minerals, organic matter, and water [6]. 

This study aims to develop an approach to reduce the concentration of heavy metal pollutants in soils and decrease the toxicity of residual contaminants to levels that enable the reuse of the polluted soil. We further evaluate the influence of various parameters on the removal of petroleum mixture in soil from waste disposal sites to enhance the rate of microbial biodegradation. 

## 2. Materials and Methods

### 2.1. Characterization of Soil and Oil Samples

Subsurface soil samples contaminated with heavy crude oil were collected from the contaminated fields of a petroleum company in the southeastern region of Tripoli, Libya, in August 2020, reflecting a range of hydrocarbon concentrations. Samples were placed in plastic bags, cooled, and then transferred to the PRC (Petroleum Research Center), and stored at 4 °C until analysis by the microbiological laboratory personnel. The soil was air-dried for 48 h and then sieved through a 2 mm mesh to remove gravel and stones. To obtain one homogenous sample, all samples were mixed. Biological, physical, and chemical characterizations of the soil and contaminants (petroleum hydrocarbon) were analyzed.

Soil suspensions in water were used to measure soil pH as described in [7]. Soil texture was analyzed using a Malvern Mastersizer 2000^®^ (Malvern Instruments Limited, Malvern, UK). X-ray diffraction (XRD) was employed to carry out mineralogy analysis. Soil moisture content and water holding capacity were measured as described in [8] using ISO 11465:1993 standard method [9]. To determine the total petroleum hydrocarbons (TPHs) level, a Soxhlet Extraction System (test method 35–40 °C) with dichloromethane was used to extract the petroleum products present in the soil samples.

#### 2.1.1. Measurement of Soil Organic Matter (OM) and Nutrient

Organic matter was determined by measuring weight loss ‘loss on ignition’ of heated soil sample at 360 °C. The soils were dried, sieved (2 mm), and analyzed for available potassium (K), nitrogen (N), and phosphorus (P). Available potassium in the soil was measured using a Jenway 6305 spectrophotometer (Thermo Fisher Scientific, Waltham, MA, USA), extracted by ammonium acetate at a pH of 7.0 [10]. Available nitrogen was determined after extraction with 50 mL 2.0 M KCl [11]. Available phosphorus was measured spectrophotometrically according to [12]. 

#### 2.1.2. Isolation of Hydrocarbon-Degrading Bacteria

After adding one gram of a dry homogenized contaminated soil sample to 10 mL of distilled water and vortexed thoroughly, 5 mL of the supernatant was used as an inoculum for the first enrichment flask that contains 50 mL of the medium with the following composition in g/L: K_2_HPO_4_ (4.65); MgSO_4_·7H_2_O (0.2); NaH_2_PO_4_·H_2_O (1.5); KCl (0.1); (NH_4_)_2_SO_4_ (1.5); casamino acid (0.5); yeast extract (0.5); peptone (0.5); and 2 mL of trace elements solution that contained (mg/L): MnCl_2_·4H_2_O (400); CoCl_2_·6H_2_O (1); H_3_BO_2_ (2); ZnSO_4_·7H_2_O (50); CuSO_4_·5H_2_O (0.4); and Na_2_MoO_4_·2H_2_O (500) [13,14]. Then, the medium was adjusted to pH = 7.1. Cultures were incubated on orbital shakers for 14 days at 37 °C to isolate hydrocarbons, utilizing bacteria from soil contaminated with oil [15]. The two bacterial isolates were identified further. Isolated bacterial strains were used as the inoculum to assist the biotransformation of heavy metals and hydrocarbons.

#### 2.1.3. Identification of Isolates via 16S rDNA Analysis

Sigma’s GenElute Bacterial Genomic Kit protocol was used for DNA extraction from the bacterial colonies. And to amplify genomic DNA from hydrocarbon degraders, polymerase chain reaction (PCR) was used with specific universal primers (27f and 1525r) that bind at the 16S rDNA-conserved 5′ and 3′ ends of the eubacteria. PCR amplification was performed as described by the authors of [16]. All the used reagents were from Bioline, London, UK. Biometra Thermal Cycler was used for performing PCR. The PCR products’ sizes were investigated by analyzing 4 μL of the PCR reactions using 1.0% (*w/v*) agarose gel electrophoresis in Tris acetic acid EDTA buffer (TAE) for 30 min at 100 V, 0.5 × TBE, stained with ethidium bromide, and visualized under UV light using (Bio-Rad Gel Doc, Flour-S multi-imager) equipped with the QuantityOne analysis software (Bio-Rad). The amplified fragment’s size was estimated through a comparison with a 100 bp molecular-size marker, as mentioned in [17]. 

ExoSAP-IT PCR clean-up kit protocol was used for cleaning the PCR products. This protocol consists of a single pipetting step (enzyme mixture addition), an incubation at 37 °C for 15 min, then a further incubation at 80 °C for 15 min to inactivate enzyme. Following this, PCR products were sequenced using Sanger sequencing based on a big dye terminator using an ABI 3730xl DNA sequence from Genius Laboratories Ltd. (based on Newcastle University campus, Newcastle upon Tyne, UK). MEGA software version 4.0 was used for viewing the sequences. A contiguous sequence was made using forward and reverse sequences and then submitted into BLAST for comparison with known databases.

### 2.2. Experimental Design

#### 2.2.1. Microbial Inoculum Preparation 

Mixed bacterial consortia isolated from the hydrocarbon-contaminated soil samples were used in this investigation. The bacteria cultures were grown on 100 mL mineral salt medium to mid-log phase (10^8^ CFU/mL) and centrifuged, washed, and re-suspended in distilled water, and then mixed in equal proportions. The mixture culture of consortium 1 containing 2 hydrocarbon degraders was used as the bacterial inoculum (0.5 g to 100 g soil fresh weight) for the bioaugmentation, and for the combination of biostimulation and bioaugmentation treatments. 

#### 2.2.2. Soil Microcosms

The experiment tested two technologies (bioaugmentation and biostimulation) with a range of different treatments to enhance bioremediation. As it was suspected that the indigenous microbial population in the contaminated soils may be insufficient for bioremediation; urea, phosphorus (P), inorganic nutrients (NPK), isolated hydrocarbon degraders (BAS), and bacterial products mixed with NPK as the combination (BAS + NPK) were added to soil to assess their potential to stimulate bioremediation, as illustrated in Table 1. 

Two sets of identical experimental microcosms were set up. Each set of experiments consisted of 100 g of soil (dry wt) placed in 250 mL dark glass containers. A control soil was left unamended to assess natural P removal levels and to evaluate hydrocarbon transformation caused by the indigenous soil microorganisms, whilst the amended soils were mixed with commercial fertilizer to provide a C:N:P ratio of 100:10:1, and 0.5 g bacterial hydrocarbon degraders were added (consortium 1). The soil was incubated at a temperature of 30 °C with a 60% water holding capacity for 90 days. Replicate (100 g) microcosms were used for each treatment. Each experimental microcosm was sampled on days 15, 30, 60, and 90 for TPH concentration, heavy metals, toxicity, pH, and viable counts of bacteria and was monitored throughout the duration of the experiment.

### 2.3. Microbiological Analysis/Total Hydrocarbon Degraders Counts 

For the estimation of total bacterial number and hydrocarbon degraders in soil, samples were developed using two different techniques: (1) dilution plate count and (2) most probable number (MPN) of hydrocarbon degraders (96-well microtiter plate) [18]. The objective of this test was to enumerate petroleum hydrocarbon-degrading microorganisms, and further, to provide a complete count of active organisms decomposing hydrocarbons. 

### 2.4. Oil Extraction

Soxhlet Extraction System (test method 35–40 °C) with dichloromethane was used to extract the petroleum products in soil samples. After extraction, the solvent evaporated. The washed anhydrous sodium sulfate solvent was used to dry out the extracted solution using Whatman filter paper number 50. Extracted hydrocarbon samples were analyzed using gas chromatography (Varian CP-3800) to measure the total petroleum hydrocarbon (TPH) level.

### 2.5. Heavy Metals Analysis

The heavy metals were monitored during the experiment. Soil samples were digested with HNO_3_ acid for heavy metal quantification analysis according to EPA 3050 B using Inductively Coupled Plasma Optical Emission Spectrometer (ICP-OES), VISTA-PRO CCD Simultaneous ICP-OES, Varian, prior, during, and after the treatment. Metals reduction by bacterial cells was calculated as the ratio of ions removal:R % = (A − B)/A × 100%(1)

### 2.6. Eco-Toxicological Test

Using a Microtox M500 analyzer, Eco-toxicity tests were carried out according to the Microtox Manual (1992). This method depends on analyzing the reduction in light emission of luminescent bacteria (Vibrio fischeri) under toxic stress. The objective of this test was to provide a useful complement to chemical analysis for evaluation of the impact of the oil contamination on toxicity in soil. 

The inhibition assay is based on the analysis of light emission reduction in luminescent bacteria Vibrio fischeri (NRRL B-111777) when exposed to a contaminated environmental sample. In this work, soil extract was exposed to V. fischeri: a soil ethanol extract was carried out on days 0, 15, 30, 60, and 90 of the bioremediation experiment. 

Soil ethanol extract method: 2 g dry wt soil sample was placed in 10 mL polypropylene centrifuge tubes and 2 mL of 95% ethanol was added (in triplicate). They were mixed for 30 min to extract all the toxic components and then centrifuged at 6000 rpm for 5 min. The supernatant was removed, and the extract was mixed with 3% sterile solution of NaCl (3% ethanol in 2% NaCl) following standard extraction procedure described in [19]. 

Freeze-dried, bioluminescent cells of V. fischeri (as supplied by the manufacturer: SDIX Europe Ltd., Southampton, UK) were resuscitated by adding 1 mL 0.1 M sterile KCl and shaking at 25 °C, 200 rpm for 1 hr. In general, assays were performed as described by the authors of [20]. After resuscitation, V. fisheri was used immediately. A total volume of 100 μL aliquot of this suspension was added to 900 μL of extracted soil ethanol solution and mixed at 15 s intervals. The luminescence of the samples was measured after 15 min of exposure using the Microtox acute toxicity test Microtox Model 500 Analyzer (SDIX Europe Ltd.), as per the manufacturer’s instructions. Microtox reagents and test solutions were also supplied by Strategic Diagnostics Inc. Three independent replicates of each assay were performed for each soil extract and the luminescence inhibition after 15 min exposure to each sample was taken as endpoint and recorded as relative light units (RLUs). 

Light emission was measured as an RLU and calculated from the mean of the three replicates for each sample as % bioluminescence relative to control samples (ethanol blank) for each assay [19,21]. Bioassays were carried out at room temperature, which varied from 18 to 20 °C. 

The toxicity effect of different concentrations of nutrient amendments were calculated by the following: % INH = Be − Se/Be × 100(2)

Or
INH (%) = (1 − (Se/Be)) × 100(3)
where Be is the emission of the blank and Se that of the sample at the different times. The INH (inhibition efficiency) values are the averages of at least three measurements, and EC50 values are calculated corresponding to INH (%) = 50. If the inhibition caused by the extracted sample is below 20%, the number of toxicants is less than the detection limit. If the inhibition is between 20 and 50%, the sample contains low amounts of toxicants, and if over 50% inhibition is observed, the sample contains high amount of toxicants [19,22,23].

### 2.7. Statistical Analysis

Statistical Package for the Social Sciences (IBM SPSS Statics 24) was used for data analysis. *t*-tests were performed to compare the means between two data groups, and *p* values < 0.05 were defined as statistically significant. Microsoft Excel (V.2202) was used to create the figures.

## 3. Results

### 3.1. Soil Chemical and Physical Analyses

The soils used in the bioremediation experiments were contaminated by petroleum hydrocarbons with a concentration of 6571.08 ppm. Table 2 illustrates the results of the soil’s physical and chemical properties, which showed low nutrient contents (the available N, P, and K were 0.051 ppm, 44.25 ppm, and 66.24 ppm, respectively), with a neutral pH of about 7.64. The total bacterial count was 2.4 × 10^6^ CFU/g soil and the fungi count were 6.9 × 10^5^ CFU/g soil, whereas the water holding capacity (WHC) was low at 4.39%, which indicates that the soil had low water storage capacity and moisture content, which is characteristic of the high evaporation nature of arid regions. 

In terms of heavy metals concentration, arsenic (As), barium (Ba), chrome (Cr), mercury (Hg), lead (Pb), nickel (Ni), and zinc (Zi) showed high concentrations in soil at 19.72, 48.01, 105.06, 166.21, 205.32, 121.77, and 234.14 ppm, respectively. However, cadmium (Cd) concentration was <0.002. 

The soil samples’ mineral compositions were also analyzed using XRDb and showed that the samples contained approximately 58.04, 11.46, and 0.50% of lepidocrocite, microcline, and wustite, respectively (Table 3).

### 3.2. Molecular-Based Identification of Bacterial Isolates

The isolates that were selected for the study are the isolates that could grow on heavy metals and showed a maximum growth. The characterization level of the genus of both bacterial isolates was determined based on their 16S ribosomal RNA gene complete sequences. 16S rRNA sequences were submitted to BLAST under accession numbers CP038030.2 and EF599311.1, and the bacteria identified are shown in Table 4.

### 3.3. Changes in pH 

pH changed during the experiment for all treatments. In control samples, the pH started at 7.5 ± 0.1 and then decreased to 7.1 ± 0.2 by the end of the experiment. With NPK treatment, the pH also started at 7.6 ± 0.2 and then decreased to 7.1 ± 0.1. In contrast, for the treatment using BAS (bacteria products), the pH started at 7.4 ± 0.2 and then decreased to 6.8 ± 0.2; similar results were obtained with treatment using urea, as shown in Table 5.

### 3.4. Microbial Counts 

Plate count technique was used to estimate viable bacteria population at intervals over 90 days. The results are shown as CFU/g, as shown in Figure 1. The initial number of indigenous bacteria in the control and soil amended with urea, P, NPK, BAS, and BAS + NPK (nutrient mixed with bacteria inocula) were in the range of (2.40 × 10^6^ CFU/g soil and 3.65 × 10^6^ CFU/g soil). Total numbers slightly increased at day 15, 30, and 60, and then slightly declined by day 90. 

### 3.5. Hydrocarbon Removal

In all treatments, the TPH concentration was measured at the start and during the experiment as well as at the end of the experiment. As shown in Table 6 and Figure 2, control soil microcosms (without treatment) have shown a slight decrease in TPH, with only 31% of TPH removed during the period of the bioremediation. However, treatment nutrients mixed with bacteria products showed the highest TPH reduction (73%), while treatment with urea and NPK presented around 43% and 45% of TPH reduction, respectively, and BAS samples (bacteria products) showed a 61% TPH reduction. The results showed that there was a clear transformation of TPHs in all samples, yet fewer levels of degradation were recorded for the control soil and for phosphorus (P) nutrient-amended soil, compared with the urea-, NPK-, BAS (bacteria products)-, and BAS + NPK (nutrient mixed with bacteria products)-amended soils.

### 3.6. Gas Chromatography Analysis of the Effective Nutrient Amendment (Urea, P, NPK) and Bacterial Products

The oil biodegradation process in control untreated soil and treatments with urea, P, NPK, BAS, and BAS + NPK were followed by evaluating the changes in the hydrocarbon content of petroleum hydrocarbons via gas chromatography (GC). Changes in TPH concentrations are presented in Figure 3 and Figure 4, respectively, as percentages remaining at 15 and 90 days compared to those at the initiation of the experiment (zero time) in all soil treatments. A GC analysis of petroleum hydrocarbons indicated that the removal of petroleum hydrocarbons varied between the five treatments, and the addition of urea, NPK, and bacterial products enhanced the degradation of medium- and long-chain compounds compared with the control.

The control samples, to which no nutrients were added, showed losses in the TPH, mainly in the shorter fraction, and were less than observed in the treated soils with BAS + NPK. After 90 days of study, a transformation of different molecular-weight compounds and the complete removal of long-chain compounds (C20–C25) with treatments using urea, BAS, and NPK + BAS were observed, whereas a low reduction in hydrocarbon content was observed in amended soils with P and NPK. The long chains of n-alkanes (C20–C25) tend to be recalcitrant due to their poor water solubility and lower bioavailability. Overall, BAS + NPK treatment was more effective in transforming the petroleum hydrocarbons than NPK alone, while BAS alone was more promising in growing and transforming the petroleum hydrocarbons. 

### 3.7. Eco-Toxicological Assay

The reductions in light emission by the bacterium were measured to indicate exposure to toxic compounds in the soil from several samples taken during the bioremediation experiment. The luminescence of ethanol extracts of oil-contaminated bioremediation samples was compared to that of the control soil and expressed as percent inhibition. The results of the toxicity tests are presented as averages (I%) for three replicates and are summarized in Figure 5. The biotoxicity analyses of the control, urea, P, NPK, BAS, and BAS + NPK performed during the 90 days of treatment showed an overall decrease throughout the study. Overall, a significant reduction (*p*-value < 0.05) in toxicity was observed with BAS and BAS +NPK (around 44–50% inhibition observed). 

### 3.8. Heavy Metals Analysis

In the original contaminated soil, heavy metal concentrations exceeded soil leachability standards. The contaminated soil at the start of the experiment was heavily contaminated with lead (Pb), nickel (Ni), zinc (Zn), chrome (Cr), and mercury (Hg) with lower levels of barium (Ba) and arsenic (As), as shown in Table 7. On the other hand, cadmium (Cd) contamination in the soil was below the acceptable value by a USEPA of 1 μg mL^−1^.

Table 8 shows the effect of each treatment on the concentrations of heavy metals. It can be noticed these treatments underwent a large reduction with all the selected heavy metals. After 90 days of experimentation, most of the heavy metal quantity recorded a drastic reduction, including the control soil sample. It can be noticed that reduction was achieved 100% in all treatments by the end of the experiment. Moreover, a high sorption capacity on heavy metals has been shown in the treatment with urea, and there was a significant reduction (*p* < 0.05) in Cr, Cu, and Pb. In addition, treatment with NPK was also very effective against Pb, Ni, Zn, As, and Ba, while bio-augmented soil (BAS) was more effective in arsenic, barium, lead, copper, and zinc bioremediation. However, the combination of bacterial products with NPK shows a further significant reduction in the zinc (Zn) concentration value (*p* < 0.05). A similar trend was also observed with mercury (Hg). 

## 4. Discussions

### 4.1. The Influence of pH Values and Organic Matter (OM) Species

Soil pH strongly influences microbial activity and should be maintained between 7 and 9. As shown in Table 4, pH values were neutral or near neutrality to slightly alkaline throughout the experiment, which should encourage bioremediation, as there is evidence that the overall rate of hydrocarbon biodegradation is higher under slightly alkaline than under acidic conditions [24,25,26]. The decrease in pH afterward may have occurred because of the production of CO_2_ due to microbial activity. 

The key parameter that controls heavy metal transfer behavior in sediment is the pH [27]. Because of the degradation of organic matter and the oxidation of acid volatile sulfide, the sediment pH usually decreases from neutral at the start to slightly acidic. This decrease occurs because of the competition between the dissolved metals and H+ for anions (e.g., OH^−^, PO_4_, CO_3_^2−^, Cl^−^, S^2−^, and SO_4_^2−^). It subsequently decreases the metals’ adsorption abilities and bioavailability, and then increases heavy metals’ mobility [28,29].

In sediment, the mobility of heavy metals can be usually determined by the solubility of organic matter. Typically, metal ions complexation with soluble organic compounds can strongly decrease their mobility; on the other hand, soluble metal complexes formation with dissolved organic compounds would increase their mobility [29]. Since heavy metals cannot be biodegraded, the remediation of heavy metal-polluted soil can be indirectly enhanced by biostimulation through the alteration of soil pH. It is widely known that soil pH can be decreased by the addition of organic materials [30], and this subsequently increases heavy metals’ solubility and hence bioavailability, which can then be easily extracted from the soil [31].

### 4.2. Effect of Nutrients Addition

Microbial growth and metabolic activity require essential macronutrients such as nitrogen and phosphorus; however, soils contaminated with hydrocarbons are often nutrient limited [32]. In this study, the local C:N:P ratios of soil 100:10:1 were optimal to enhance TPH removal. The relationship observed showed that there was a positive influence of nutrient types and levels of microorganisms as shown in Figure 1.

In this study, the results obtained with urea-amended soils showed that the bacteria count increased to 1.23 × 10^7^ CFU/g soil at 60 days and decreased to 0.79 × 10^6^ CFU/g soil by the end of the experiment. The addition of NPK increased bacteria counts to 2.57 × 10^7^ CFU/g soil at 60 days and decreased to 8.73 × 10^6^ CFU/g soil. The number of microorganisms tends to rise during biostimulation, so during bioremediation, the increase in the number of degrading bacteria can be used as potential bioindicators as was mentioned in [33].

The rate of TPH biodegradation is effectively increased by the addition of nutrients. Treatment using urea and NPK showed a TPH transformation during the experiment, in which 43% and 45% of total petroleum hydrocarbons were removed, respectively. Because of the high nitrogen content and the relatively slow nitrogen-releasing characteristics of urea, it is considered to be a popular nitrogen source for enhanced biodegradation [34]. 

Control soils demonstrated a low reduction in heavy metals, which might have occurred due to the presence of naturally occurring heavy metals available in an insoluble form that is not readily available for microorganisms, and this can be improved by adding nutrients or/and the bio-augmentation of indigenous bacteria.

### 4.3. Effect of Bacterial Products Addition

In this work, the use of mixed microbial inoculum improved the level of petroleum removal. The control soil microcosms (without treatment) have shown a 31% decrease in TPHs during the period of the bioremediation. While the bacteria-augmented soil had a higher TPH degradation efficiency than the control untreated soil with about 61% of TPH removed after 90 days. The numbers of indigenous bacteria increased from 2.4 × 10^6^ and 2.7 × 10^6^ to 4.15 × 10^7^ and 3.1 × 10^7^ CFU/g soil at 30 days, respectively. The bioaugmentation proved to be a better approach as shown in Figure 2. High TPH reduction reached more than 70% in 12 weeks of incubation of contaminated soil. A lower degradation percentage of (58%) was achieved with the same soil before bioaugmentation.

Although most of the microbe-assisted remediation is performed ex situ, an extremely vital in situ microbe-assisted remediation is the soluble mercuric ions Hg (II) microbial reduction to volatile metallic mercury and Hg (0), for which mercury-resistant bacteria are responsible [27,35]. The reduced Hg (0) then can easily volatilize out of the environment. 

### 4.4. Effect of Treatments on Toxicity Reduction

The percentage of inhibition recorded from acute toxicity tests was expressed with respect to the treated samples. The toxicity of the control soil, urea, and P-treated soils remained quite high throughout the experiment. On the other hand, treatments with NPK and bacterial products presented the highest reduction in toxicity (44–50% reduction observed) as shown in Figure 5. There was a good correlation between the light inhibition percentage obtained from this assay and TPH reduction, probably because both bacterial products and the nutrient type and the used ratio represented the right amendment needed. 

Bioprecipitation is one of the indirect ways by which bioremediation is performed with sulfate-reducing bacteria (Desulfovibrio desulfuricans). This bacteria can produce hydrogen sulfate from sulfate, which subsequently reacts with heavy metals such as Cd and Zn to form insoluble metal sulfides [36].

In situ, pollutant metals within contaminated soils could become immobilized by converting them into their metal phosphates, consequently reducing their bioavailability [37]. For instance, many metal phosphates (e.g., Pb, Zn, Cd) are highly insoluble [38].

The adoption of remediation technology for sediment contaminated by heavy metal generally depends on some sediment special characteristics, such as size distributions of particles, metal species distribution, and metal loads [29]. Many microorganisms, especially bacteria (Pseudomonas putida, Bacillus subtilis, and Enterobacter cloacae), have been successfully used to reduce Cr (VI) to less toxic Cr (III) [39,40]. Also, some studies reported that Bacillus subtilis has the ability to lessen non-mineral elements. For example, [41] reported that Bacillus subtilis converted selenite to the less toxic elemental selenium.

## 5. Conclusions

Preliminary laboratory work undertaken in the first phase of this research established that hydrocarbon-degrading microbes were present in the contaminated soil samples and small-scale laboratory experiments indicated that nutrient addition (N and P) stimulated the microbial-based reduction in hydrocarbons presents in the soil. The study compared the effectiveness of microbial inocula and nutrient addition on the biotransformation of TPH-contaminated soil. The introduced inoculum achieved a >60% reduction in hydrocarbon pollutants. Nutrients addition alone was also effective in improving the bioremediation compared with the untreated control soil. The samples treated with bacterial inocula alone showed a high level of TPH transformation (61%), indicating that the indigenous bacterial population present has excellent TPH transformation capacity. The bio-augmentation of polluted soils is a potentially positive method to enhance indigenous soil microbes to reduce hydrocarbon contents and remediate the environment.

Overall, a 73% TPH reduction was achieved using bacterial products mixed with nutrient amendment (N and P). The bio-augmented soil and nutrient-amended soil displayed high potential bioremediation levels and showed the highest reduction in lead, arsenic, and barium concentrations. The toxicity of the contaminant (petroleum hydrocarbons containing heavy metals) was also reduced by 50%. However, metal reduction may be due to their mobilization to lower soil strata. In conclusion, these investigations demonstrated the potential of nutrients addition and bacterial products as an environmentally attractive approach for hydrocarbon remediation. A proper soil management strategy, however, should be implemented to prevent the percolation of toxic metals into ground waters and aquifer bodies in the area.

## Figures and Tables

**Figure 1 microorganisms-11-02577-f001:**
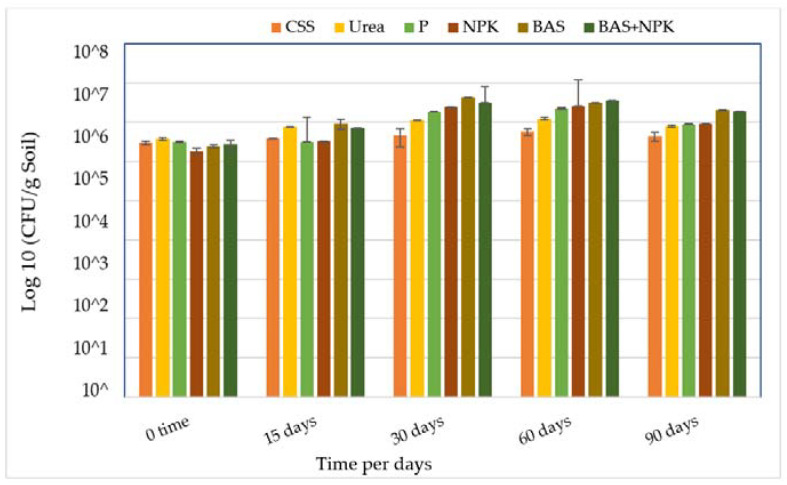
Changes in hydrocarbon degrader counts (CFU g^−1^ soil) in contaminated soil samples with treatments. Bars represent the average of triplicate plate counts; error bars indicate the standard deviation of the triplicates. Abbreviations: CSS, control soil sample; P, phosphorus; NPK, nitrogen, phosphorous, and potassium; BAS: bio-augmentation soil.

**Figure 2 microorganisms-11-02577-f002:**
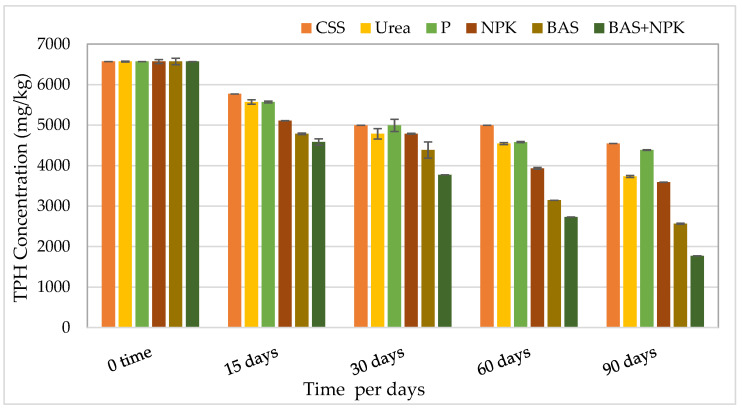
Reduction in TPHs in contaminated soil samples with treatments. Bars represent the average of triplicate microcosms readings; error bars indicate the standard deviation of the triplicates. Abbreviations: Abbreviations: CSS, control soil sample; P, phosphorus; NPK, nitrogen, phosphorous, and potassium; BAS: bio-augmentation soil.

**Figure 3 microorganisms-11-02577-f003:**
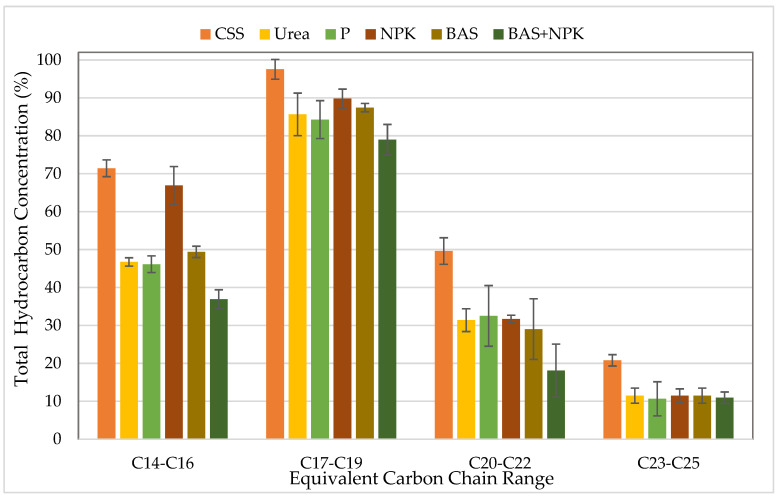
Representative GC chromatograms of TPHs extracted from petroleum-contaminated soil after 15 days of incubation, shown as percentages remaining compared to those at the initiation of the experiment (zero time). Bars represent the average of triplicate samples. Error bars indicate the standard deviation of the triplicates. Abbreviations: CSS, control soil sample; P, phosphorus; NPK, nitrogen, phosphorous, and potassium; BAS: bio-augmentation soil.

**Figure 4 microorganisms-11-02577-f004:**
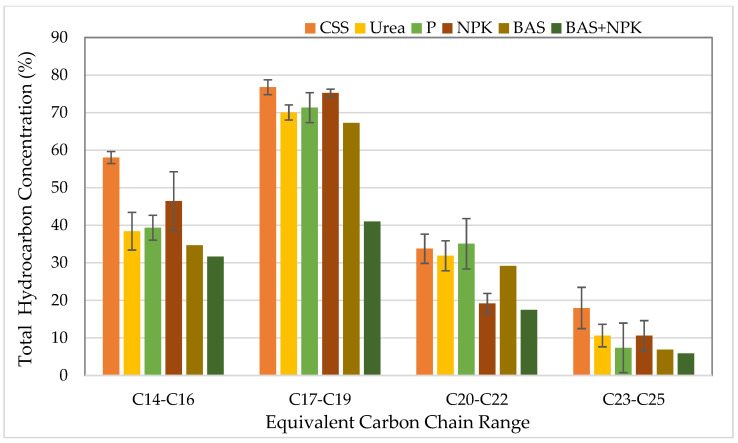
Representative GC chromatograms of TPHs extracted from petroleum-contaminated soil after 90 days of incubation shown as percentages remaining compared to those at the initiation of the experiment (zero time). Bars represent the means of triplicate samples. Error bars indicate the standard deviation of the triplicates. Abbreviations: CSS, control soil sample; P, phosphorus; NPK, nitrogen, phosphorous, and potassium; BAS: bio-augmentation soil.

**Figure 5 microorganisms-11-02577-f005:**
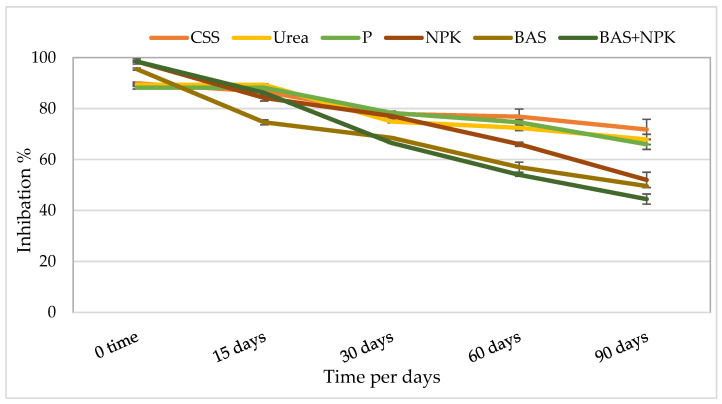
Biotoxicity values (acute toxicity assay) during bioremediation treatment determined by *V. fisheri bacteria*.

**Table 1 microorganisms-11-02577-t001:** Supplements added to the different treatments at the beginning of the experiments.

Treatments	Supplements Added
Treatment 1	None (control)
Treatment 2	Urea
Treatment 3	Phosphorous
Treatment 4	Agricultural fertilizers (N.P.K)
Treatment 5	Bacterial inoculum (consortium 1)
Treatment 6	Equal mixtures of N.P.K, bacterial inoculum (consortium 1)

**Table 2 microorganisms-11-02577-t002:** General physical and chemical properties of dry contaminated soil.

Contaminated Soil Properties	
Sampling location	Southeastern region of Tripoli
Texture	Sandy
pH	7.64
Water holding capacity (%)	4.39
Available C (ppm)	1.58
Available N (ppm)	0.051
Available P (ppm)	44.25
Available K (ppm)	66.24
TPH (mg/kg)	6571.08
CFU/g soil bacteria	2.40 × 10^6^
CFU/g soil fungi	6.9 × 10^5^
As (ppm)	19.72
Ba (ppm)	48.01
Cd (ppm)	<0.002
Total Cr (ppm)	105.06
Total Hg (ppm)	166.21
Pb (ppm)	205.32
Ni (ppm)	121.77
Zn (ppm)	234.14
Hg (ppb)	5.83

**Table 3 microorganisms-11-02577-t003:** Minerology composition of soil sample using XRD analysis.

Ref. Code	Compound Name	Chemical Name	Chemical Formula	Approximate %
44-1415	Lepidocrocite	Iron Oxide Hydroxide	FeO(OH)	58.04
19-0932	Microcline	Potassium Aluminum Silicate	KAISi_3_O_8_	41.46
06-0615	Wustite	Iron Oxide	FeO	0.50

**Table 4 microorganisms-11-02577-t004:** Identification of selected hydrocarbon-degrading bacterial isolates via 16S rDNA gene sequencing.

Isolates	Species as Close Relatives	Accession No.	Similarity (%)
HKI	*Rhodococcus ruber*	CP038030.2	100.0
HKII	*Beta proteobacterium*	EF599311.1	99.1

**Table 5 microorganisms-11-02577-t005:** Changes in soil pH during the experiment.

* Treatments	pH (In Double Distilled H_2_O)				
	0 Day	15 Days	30 Days	60 Days	90 Days
CSS	7.5 ± 0.1	7.1 ± 0.1	6.6 ± 0.2	7.0 ± 0.2	7.1 ± 0.2
UREA	7.8 ± 0.0	7.9 ± 0.3	7.7 ± 0.2	7.2 ± 0.1	6.8 ± 0.2
P	7.7 ± 0.1	7.9 ± 0.2	7.5 ± 0.1	6.7 ± 0.2	7.6 ± 0.1
NPK	7.6 ± 0.2	8.2 ± 0.1	7.3 ± 0.2	7.1 ± 0.1	7.1 ± 0.1
BAS	7.4 ± 0.2	8.1 ± 0.2	8.3 ± 0.3	7.1 ± 0.1	6.8 ± 0.2
BAS + NPK	7.7 ± 0.1	8.4 ± 0.3	7.9 ± 0.1	6.4 ± 0.1	7.2 ± 0.1

* CSS: Control soil sample; NPK: nitrogen, phosphorous, and potassium; BAS: bio-augmentation soil.

**Table 6 microorganisms-11-02577-t006:** TPH reduction in contaminated soil samples with treatments.

Treatments	0 Time	15 Days	30 Days	60 Days	90 Days
CSS	0%	12%	24%	24%	31%
Urea	0%	15%	27%	30%	43%
P	0%	12%	24%	31%	33%
NPK	0%	22%	27%	40%	45%
BAS	0%	27%	33%	52%	61%
BAS + NPK	0%	30%	43%	58%	73%

**Table 7 microorganisms-11-02577-t007:** Heavy metal quantity of five selected soil treatments (urea, P, NPK, BAS, BAS + NPK) measured in μg mL^−1^ at the start of the experiment.

Treatments	As	Ba	Cd	Cr	Cu	Pb	Ni	Zn	Hg
Contaminated soil samples (PPM)	56.95	48.01	<0.002	105.06	166.21	205.32	121.77	234.14	5.83 ppb

**Table 8 microorganisms-11-02577-t008:** Heavy metal quantity of five selected soil treatments (urea, P, NPK, BAS, BAS + NPK) measured in μg mL^−1^ at 90 days.

Treatments	As	Ba	Cd	Cr	Cu	Pb	Ni	Zn	Hg
CSS	37.29	45.59	<0.002	91.18	144.99	178.27	87.78	150.86	5.37
Urea	38.57	41.94	<0.002	67.85	94.74	147.52	62.23	145.55	3.94
P	31.2	38.01	<0.002	85.6	108.27	166.21	73.72	143.36	3.94
NPK	19.72	33.59	<0.002	69.47	128.2	142.27	65.45	145	4.33
BAS	20.59	32.35	<0.002	72.91	133.19	142.34	96.77	139.25	4.13
BAS + NPK	13.35	26.51	<0.002	66.87	103.9	142.34	74.61	125.79	2.86

## Data Availability

The datasets generated during and/or analyzed during the current study are available from the corresponding author upon reasonable request.

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
