# Peer review of "Remediation Approaches to Reduce Hydrocarbon Contamination in Petroleum-Polluted Soil"

_microorganisms, 2023, doi:10.3390/microorganisms11102577_

Round 1
Reviewer 1 Report
The remediation of hydrocarbon and toxic metals in petroleum polluted soils are of high importance. The authors approach to employ augmentation of bacteria with additional nutrients to stimulate the biotransformation of contaminated soil was investigated, and the authors showed significant reduction of total petroleum hydrocarbon and metals in polluted soil. The research is valuable and has practical applications to reduce environmental pollution.
Some comments are mentioned in the attached manuscript. None of the graphs and tables were marked that results differ significantly while the text states that. Please mark the significant differences where applicable.

The revision of English language is required since it is not clear why some words begin with capital letters, e.g. lead, Zinc and Barium?
Author Response
Response to Reviewer X Comments
- Summary
Thank you very much for taking the time to review this manuscript. Please find the detailed responses below and the corresponding corrections highlighted in the re-submitted files.
2. Questions for General Evaluation
|
Reviewer’s Evaluation |
Response and Revisions |
Does the introduction provide sufficient background and include all relevant references?
|
Yes/Can be improved/Must be improved/Not applicable |
|
Are all the cited references relevant to the research?
|
Yes/Can be improved/Must be improved/Not applicable |
|
Is the research design appropriate?
|
Yes/Can be improved/Must be improved/Not applicable |
|
Are the methods adequately described?
|
Yes/Can be improved/Must be improved/Not applicable |
|
Are the results clearly presented? |
Yes/Can be improved/Must be improved/Not applicable |
|
Are the conclusions supported by the results?
|
Yes/Can be improved/Must be improved/Not applicable |
|
- Point-by-point response to Comments and Suggestions for Authors
Comments 1: [Capital letters]
Response 1: [nitrogen, potassium, and phosphorus]. Thank you for pointing this out. We agree with this comment. Therefore, we have changed these words to start with lowercase letters. This change can be found – Page 1, abstract, line21.
Comments 2: [Some metals (lead) are in lowercase letters while zinc and barium are in capital letters]
Response 2: Agree. We have, accordingly, changed all these words in the manuscript to be in lowercase letters [lead, zinc, and barium], this can be found – (Page 1, abstract, line23) and (page 6, paragraph 3.1, line 224).
Comments 3: [Metals do not break down please correct the statement]
Response 3: [which is essential to transform and detoxify heavy metals in the environment]
Thank you for pointing this out. We agree with this comment. Therefore, we have changed the word breakdown to more appropriate words (transform and detoxify). This change can be found – page 1, introduction, line 34].
Comments 4: [Hg is toxic in all forms: metallic, organic, and inorganic. Please use appropriate terms]
Response 4: Agree. We have, accordingly, changed the sentence [Heavy metals are toxic in their chemically combined forms (as with mercury)] to [Heavy metals are toxic in their chemically combined forms, some are extremely toxic in most of its forms (as with mercury), and others are toxic in their elemental form] this change can be found – page 2, paragraph 2 in the introduction, line 43].
Comments 5: [Please refer to earlier comment is duplication of the statement (row 34-360)].
Response 5: Agree. We have, accordingly, changed the sentence [In some instances, a chelating agent may be necessary to break down the toxic effects of heavy metals.] to [In some instances, a chelating agent may be essential to treat the toxic effects of heavy metals.] this change can be found – page 2, paragraph 3 in the introduction, line 53].
Comments 6: [A lowercase letter].
Response 6: Agree. [physical, and chemical characterization] this change can be found – page 2, paragraph 2.1, line 69].
Comments 7: [lowercase letter vs. capital letters].
Response 7: [Measurement of soil organic matter (OM) and nutrient] this change can be found – page 2, paragraph 2.1.1, line 78].
Comments 8: [lowercase letter vs. capital letters].
Response 8: [adding one gram of a dry homogenized] this change can be found – page 3, paragraph 2.1.2, line 87].
Comments 9: [What kind of trace elements the solution contained].
Response 9: [2 ml of trace Elements solution that contained (Gerhardt, 1981; Plaza, 2008) (mg/L): MnCl2.4H2O (400); CoCl2.6H2O (1); H3BO2 (2); ZnSO4.7H2O (50); CuSO4.5H2O (0.4); and Na2MoO4.2H2O (500)] Thank you for pointing this out. We agree with this comment. Therefore, we have added the composition of trace elements that we used in this study. This change can be found – page 3, paragraph 2.1.1, line 91-93].
Comments 10 and 11: [Total chromium? total mercury?].
Response 10 and 11: [Total Cr, total Hg]. These changes can be found – page 6, table 2].
Comments 12: [lowercase letter vs. capital letters].
Response 12: [lepidocrocite, microcline, and wustite]. This change can be found – page 6, paragraph 3.1, line229-230].
Comments 13: [Do heavy metals grow?].
Response 13: [could grow on heavy metals]. Thank you for pointing this out. A simple mistake can change the whole meaning. This change can be found – page 7, paragraph 3.2, line 233].
Comments 14: [Please specify the toxicity of what was reduced by 50%].
Response 14: [The toxicity of the contaminant (petroleum hydrocarbons containing heavy metals) was also reduced by 50%]. To explain this point, we have added the specific pollutant toxicity that was reduced by 50 % which is petroleum hydrocarbons containing heavy metals. This change can be found – page 16, Conclusions, line 446-447].
- Response to Comments on the Quality of English Language
Point 1:
Response 1: (We have revised the manuscript and corrected all the words with some mistakes)

Reviewer 2 Report
The experimental part is very poorly written, hard to follow. Also, the methods are not well explained. It is necessary to perform an ecotoxicity test for the soil and not for the water system. The comments are included in the paper.
Author Response
Response to Reviewer X Comments
- Summary
Thank you very much for taking the time to review this manuscript. Please find the detailed responses below and the corresponding corrections highlighted in the re-submitted files.
2. Questions for General Evaluation |
Reviewer’s Evaluation |
Response and Revisions |
Does the introduction provide sufficient background and include all relevant references? |
Yes/Can be improved/Must be improved/Not applicable |
|
Are all the cited references relevant to the research? |
Yes/Can be improved/Must be improved/Not applicable |
|
Is the research design appropriate? |
Yes/Can be improved/Must be improved/Not applicable |
|
Are the methods adequately described?
|
Yes/Can be improved/Must be improved/Not applicable |
|
Are the results clearly presented? |
Yes/Can be improved/Must be improved/Not applicable |
|
Are the conclusions supported by the results?
|
Yes/Can be improved/Must be improved/Not applicable |
|
- Point-by-point response to Comments and Suggestions for Authors
Comments 1: [Low letters]
Response 1: [(nitrogen, potassium, and phosphorus) and (zinc, and barium)]. Thank you for pointing this out. We agree with this comment. Therefore, we have changed these words to start with lowercase letters. This change can be found – Page 1, abstract, line 21and 23.
Comments 2: [Describe how you sampled the soil and how you stored soil until it was analysed]
Response 2: [Subsurface soil samples contaminated with heavy crude oil were collected from Petroleum Company-contaminated fields East-Southern Region of Tripoli, Libya, in August 2020, to reflect a range of hydrocarbon concentrations. Samples were placed in plastic bags and cooled then transferred to the PRC (Petroleum research Center) and stored at 4 ◦C until analysis by the microbiological laboratory.]. We explained how the samples were collected and stored. This change can be found – (Page 2, paragraph 2.1, line63-67).
Comments 3: [Which contaminants?]
Response 3: [contaminants (petroleum hydrocarbon)]
Thank you for pointing this out. We have described the type of pollutant, and this change can be found – (Page 2, paragraph 2.1, line70].
Comments 4: [Please add standard method]
Response 4: [Soil moisture content and water holding capacity were measured as described by [15] using ISO 11465:1993 standard method]. We have, accordingly, added the standard method of soil moisture content. This change can be found – page 2, paragraph 2.1, line 74-75].
Comments 5: [Please describe how did you prepared soil for analyses].
Response 5: Subsurface soil samples contaminated with heavy crude oil were collected from Petroleum Company-contaminated fields East-Southern Region of Tripoli, Libya, in August 2020, to reflect a range of hydrocarbon concentrations. Samples were placed in plastic bags and cooled then transferred to the PRC (Petroleum research Center), and stored at 4 ◦C until analysis by the microbiological laboratory.
Comments 6: [But how did you determined N? which N, total N?].
Response 6: [Available nitrogen was determined after extraction with 50ml 2.0 M KCl [15]. This change can be found – page 3, paragraph 2.1.1, line 83-84].
Comments 7: [Dry or wet? This is very important information].
Response 7: [One gram of a dry homogenized contaminated soil sample]. This change can be found – page 2, paragraph 2.1.2, line 87].
Comments 8: [Please put this in table, it is very difficult to follow where is heavy metals?].
Response 8:
[Table1: supplements added to the different treatments at the beginning of the experiments
Treatments |
Supplements added
|
Treatment 1 |
None (Control) |
Treatment 2 |
Urea |
Treatment 3 |
Phosphours |
Treatment 4 |
Agricultural fertilizers (N.P.K) |
Treatment 5 |
Bacterial inoculum (Consortium 1) |
Treatment 6 |
equal mixtures of N.P.K, bacterial inoculum (consortium 1) |
Two sets of identical experimental microcosms were set up. Each set of experiments consisted of 100 g of soil (dry wt) placed in 250 ml dark glass containers. A control soil was left unamended to assess natural P removal levels and to evaluate hydrocarbon transformation caused by the indigenous soil microorganisms, whilst the amended soils were mixed with commercial fertilizer to give a C:N:P ratio of 100:10:1, and 0.5 g bacterial hydrocarbon degraderswere added (Consortium 1). The soil was incubated at a temperature of 30°C with a 60% water holding capacity for 90 days. Replicate (100 g) microcosms were used for each treatment. Each experimental microcosm was sampled on days 15, 30, 60, and 90 for TPH concentration, heavy metals, toxicity, pH, and viable counts of bacteria and was monitored throughout the duration of the experiment.]
Thank you for pointing this out. We agree with this comment. Therefore, we have changed the text into a table as you requested. This change can be found – page 4, paragraph 2.2.2, line 135-146].
Comments 9: [you need to have also a method for soil, this is a method for water. For this method, you have to prepare an eluate, which you have not described, and it may happen that not all substances will be leached and you will not get a good estimate of Eco toxicity. In previous works, the mentioned issue has already been explained].
Response 9: [The Inhibition assay is based on the analysis of light emission reduction of luminescent bacteria Vibrio fischeri (NRRL B-111777) when exposed to a contaminated environmental sample. In this work soil extract was exposed to V. fischeri: a soil ethanol extract was carried out on days 0, 15, 30, 60, 90 of the bioremediation experiment.
Soil ethanol extract method: 2 g dry wt soil sample was placed in 10ml polypropylene centrifuge tubes and 2ml of 95% ethanol was added (in triplicate). They were mixed for 30 minutes to extract all the toxic components and then centrifuged at 6000rpm for 5 minutes. The supernatant was removed, and the extract mixed with 3% sterile solution of NaCl (3% ethanol in 2% NaCl) following standard extraction procedure descried by [8].
Freeze dried, bioluminescent cells of V. fischeri (as supplied by the manufacturer: SDIX Europe Ltd) were resuscitated by adding 1 mL 0.1 M sterile KCl and shaking at 25 °C, 200 rpm for 1 hr. In general assays were performed as described by [26]. After resuscitation, V. fisheri was used immediately. 100 μl aliquot of this suspension was added to 900 μl of extracted soil ethanol solution and mixed at 15-s intervals. The luminescence of the samples was measured after 15 min. exposure using the Microtox acute toxicity test Microtox Model 500 Analyzer (SDIX Europe Ltd.), as per the manufacturer’s instructions. Microtox reagents and test solutions were also supplied by Strategic Diagnostics Inc. Three independent replicates of each assay were performed for each soil extract and the luminescence inhibition after 15 min exposure to each sample was taken as endpoint and recorded as relative light units (RLU).
Light emission was measured as RLU and calculated from the mean of the three replicates for each sample as % bioluminescence relative to control samples (ethanol blank) for each assay [8, 29] Bioassays were carried out at room temperature which varied from 18 to 20 ºC.
The toxicity effect of different concentrations of nutrient amendments calculated by the following
% INH = Be − Se /Be × 100
Or
INH (%) = (1-(Se /Be)) × 100
Where Be is the emission of the blank and Se that of the sample at the different times. The INH (Inhibition efficiency) values are the averages of at least three measurements, EC50 values are calculated corresponding to INH (%) = 50. If the inhibition caused by the extracted sample is below 20 %, the number of toxicants is less than the detection limit. If the inhibition is between 20 - 50 %, the sample contains low amounts of toxicants and if over 50 % inhibition is observed, the sample contains high amount of toxicants [6,8,38]]. We totally agree with this comment. Therefore, we have explained this method in detail. This change can be found – page 5-6, paragraph 2.6, line 173-207].
Comments 10: [Dry or wet?].
Response 10: [Dry contaminated soil]. This change can be found – page 6, table 2, line 223].

Round 2
Reviewer 2 Report
The paper can be accepted.